# Contribution of LRP1 in Human Congenital Heart Disease Correlates with Its Roles in the Outflow Tract and Atrioventricular Cushion Development

**DOI:** 10.3390/genes14040947

**Published:** 2023-04-21

**Authors:** Angelo B. Arrigo, Wenjuan Zhu, Kylia A. Williams, Carla Guzman-Moreno, Cecilia Lo, Jiuann-Huey I. Lin

**Affiliations:** 1Department of Developmental Biology, School of Medicine, University of Pittsburgh, 530 45th St, Pittsburgh, PA 15201, USA; 2Centre for Cardiovascular Genomics and Medicine, Chinese University of Hong Kong, Hong Kong 999077, China; 3Department of Critical Care Medicine, School of Medicine, University of Pittsburgh, 4401 Penn Avenue, Pittsburgh, PA 15224, USA; 4UPMC Children’s Hospital of Pittsburgh, 4401 Penn Ave., Pittsburgh, PA 15224, USA

**Keywords:** LRP1, congenital heart defect, outflow tract, atrioventricular cushion

## Abstract

Due to the prevalence of congenital heart disease in the human population, determining the role of variants in congenital heart disease (CHD) can give a better understanding of the cause of the disorder. A homozygous missense mutation in the LDL receptor-related protein 1 (*Lrp1*) in mice was shown to cause congenital heart defects, including atrioventricular septal defect (AVSD) and double outlet right ventricle (DORV). Integrative analysis of publicly available single-cell RNA sequencing (scRNA-seq) datasets and spatial transcriptomics of human and mouse hearts indicated that *LRP1* is predominantly expressed in mesenchymal cells and mainly located in the developing outflow tract and atrioventricular cushion. Gene burden analysis of 1922 CHD individuals versus 2602 controls with whole-exome sequencing showed a significant excess of rare damaging *LRP1* mutations in CHD (odds ratio (OR) = 2.22, *p* = 1.92 × 10^−4^), especially in conotruncal defect with OR of 2.37 (*p* = 1.77 × 10^−3^) and atrioventricular septal defect with OR of 3.14 (*p* = 0.0194). Interestingly, there is a significant relationship between those variants that have an allele frequency below 0.01% and atrioventricular septal defect, which is the phenotype observed previously in a homozygous N-ethyl-N-nitrosourea (ENU)-induced *Lrp1* mutant mouse line.

## 1. Introduction

LDL receptor-related protein 1 (LRP1) is a member of the LDL receptor family and is a multifunctional receptor that binds to multiple ligands; LRP1 plays important roles in orchestrating different cellular and molecular functions. We previously identified a homozygous missense mutation in this *Lrp1* in mice cause congenital heart defects (CHDs), including atrioventricular septal defect (AVSD) and double outlet right ventricle (DORV) [1]. We provided evidence that LRP1 function in neural cardiac crest cells is required for normal outflow tract (OFT) alignment and atrioventricular cushion (AVC) development. To explore the potential mechanism by how LRP1 mediates cardiovascular diseases, expression patterns in adult human tissues, human embryos, and mouse embryos were investigated.

To investigate the role of LRP1 in human CHD, a large patient cohort of 1922 CHD cases from the Pediatric Cardiac Genomics Consortium (PCGC) [2] and the UPMC Children’s Hospital of Pittsburgh (CHP), plus 2602 controls from the Alzheimer’s Disease Sequencing Project (ADSP) [3] were analyzed. We accessed whole-exome sequencing data and conducted the gene-burden analysis to study the association of rare putative damaging variants (PDVs) (resulting in nonsense, start-loss, splice-site, frameshift indel, non-frameshift indel, and missense disruption) in *LRP1* with CHD phenotypes. Putative damaging missense (D_Mis) was called likely damaging by at least 4 of 9 prediction algorithms. Published single-cell RNA sequencing (scRNA-seq) and spatial transcriptomics data were employed to uncover spatial-resolved and cellular expression of LRP1 in mice [4] and human embryonic hearts [5].

## 2. Materials and Methods

### 2.1. Immunostaining

Embryos were fixed in 4% paraformaldehyde (Electron Microscopy Sciences, 15710, Hartfield, PA, USA) overnight. Samples from cryosection were rinsed in Tris-Buffered Saline (TBS) (Fisher Chemical, BP2471-1, Hampton, NH, USA) with 0.1% Triton X-100 (Sigma, T9284, St. Louis, MO, USA). Samples were then blocked for 1 h at room temperature in 5% non-fat milk, 0.1% Triton X-100, TBS. Embryos were then incubated at 4 °C overnight for LRP1 (Abcam, ab92544, 1:200, Cambridge, UK) and AP2α (DHSB, 3B5, 1:25). Post antibody incubation was washed with TBS with 0.1% Tween-20 (Fisher Scientific, BP337, Hampton, NH, USA). Secondary incubation of DAPI (Thermo Scientific, 62248, Waltham, MA, USA), donkey anti-rabbit Alexa Flour 555, and donkey anti-mouse 488 took place at room temperature for an hour. Samples were rinsed three more times with TBS with 0.1% Tween-20, then mounted on SuperFrost Plus Microscope Slides (Fisher Scientific, 12-550-15, Hampton, NH, USA) with ProLong Glass Antifade Mountant (Invitrogen, P36982, Waltham, MA, USA). Embryos were then imaged using Leica TCS SP8 (Wetzlar, Germany).

### 2.2. Analysis of Publicly Available Single-Cell RNA Sequencing Data

Publicly available raw single-cell RNA sequencing (scRNA-seq) data of the mouse embryonic hearts from E8.5 to E10.5 were downloaded from the NCBI GEO database under accession number GSE76118 [4]. Expression levels were quantified using RSEM v1.3.3. We generated a cell × gene transcript per million (TPM) matrix at the gene level after aggregating the expression of all cells together. Expression of all genes in the *LRP1* cluster was combined to generate the expression of the LRP1. Downstream analyses were performed as described previously [5,6].

### 2.3. Human Study Participants

All data access requests, and human studies were approved by the Institutional Review Board of the University of Pittsburgh School of Medicine and the UPMC Children’s Hospital of Pittsburgh (CHP). All consenting participants were approved by the relevant review committees. The personal identities of the study participants were encrypted and secured in accordance with approved guidelines and regulations. This research was partly supported by the University of Pittsburgh Center for Research Computing through computing resources provided. We analyzed whole-exome sequencing (WES) data from 471 CHD patients from the UPMC Children’s Hospital of Pittsburgh UPMC (Pitt), 1451 CHD patients from the Pediatric Cardiac Genomics Consortium (PCGC) [2], and 2602 controls from the Alzheimer’s Disease Sequencing Project (ADSP) [3] with European ancestry. To investigate phenotype-specific effects, patients were grouped into those with conotruncal defect (CTD), left outflow tract obstruction (LVOTO), and atrioventricular septal defect.

### 2.4. Recovery of Rare Predicated Pathogenic LRP1 Variants

For Pitt subjects, whole-exome sequencing (WES) was carried out on Illumina HiSeq2000 (BGI genomics, Cambridge, USA) with 100 paired-end reads at 80–100× coverage using Agilent V4 or V5 exome capture kit (Agilent, Santa Clara, CA, USA). For samples obtained from the PCGC (dbGaP phs001194.v2.p2) and healthy control samples obtained from the ADSP (NG00067.v2), SRA files were downloaded from the NCBI SRA database and converted to FASTQ files using SRA-toolkit (BIOWULF, Bethesda, MD, USA). BWA-MEM [7] was used to align reads in FASTQ files to the human reference genome GRCh38. BAM files were further processed using GATK4 Best Practices workflows [8]. The intersection of the WES capture kit intervals used to sequence each cohort was taken, and single nucleotide variants (SNVs) and small indels (InDels) were detected individually using GATK HaplotypeCaller (BIOWULF, Bethesda, MD, USA) and jointly called using GATK GenotypeGVCFs (BIOWULF, Bethesda, MD, USA). Further quality filtering was applied using bcftools 1.9 [9] and qctool 2.0.6. High-quality variants were recovered that: (1) have excess heterozygosity *p*-value > 3.4 × 10^−6^; (2) passed GATK Variant Score Quality Recalibration (VSQR) with 99.95% sensitivity; (3) have SNV or indel genotype quality ≥ 20 or ≥ 60, respectively; (4) are SNVs or InDels not within 10 bp or 5 bp of an indel, respectively; (5) have missing rate < 10% and differential missingness *p*-value > 10^−6^; and (6) have control HWE *p*-value > 10^−6^. Variants were annotated using Ensembl VEP v102 [10] with variant identifiers, gene symbol in NCBI RefSeq v109 [11], a variant consequence of the most severely affected transcript, allele frequency in gnomAD exomes v2.1.1 [12], ClinVar [13] significance, and variant deleteriousness predictors such as SIFT [14] and PolyPhen [15]. Phred-scaled CADD scores [16] were obtained from CADD v1.6. Only variants identified in *LRP1* were used for this analysis. Samples with a FREEMIX score [17] greater than 0.075 were considered contaminated and removed before filtering. Samples with missingness greater than 10% and outliers in the number of variants present were removed before analysis. To remove pairs with cryptic relatedness, one sample was removed for each pair found to be related by pedigree or KING kinship analysis [18] (PLINK, cutoff = 0.09375 for second-degree relatives), and samples with 5 or more relationships were removed. Principal component analysis (PCA) was performed using genotypes of common variants with AF > 0.05 in PLINK 1.9 [19] to determine samples with European ancestry similar to CHP in PCGC and ADSP cohorts. A total of 471 Pitt cases, 1451 PCGC cases, and 2602 ADSP controls passed sample-level filtering. In addition, only protein-altering variants were retained for analysis, including predicted loss-of-function (LoF) mutations (nonsense, canonical splice-site, frameshift indels, and start loss), inframe indels, and predicted damaging missense mutation (D_Mis). As many missense variants are tolerant and would affect the degree of enrichment of pathogenic variants in a case-cohort, only predicted D_Mis called likely pathogenic by at least 4 of 9 prediction algorithms (SIFT, Polyphen2_HDIV, LRT, MutationTaster, MutationAssessor, FATHMM, PROVEAN, MetaSVM, M_CAP) were kept for downstream analyses (Appendix A).

### 2.5. Gene-Based Burden Testing

The Genome Aggregation Database (gnomAD) exome v2.1.1 (125,748 exomes) or ExAC (60,706 exomes) databases integrating large-scale exome sequencing projects with variable ancestry backgrounds were used as controls for gene burden analysis, similar to our previous published studies [6]. We tested whether there is a significant excess of *LRP1* rare damaging variants in a case-cohort compared to the control cohort (gnomAD exome v2.1.1) using only high-confidence pathogenic variants as described above. For the control cohort (gnomAD exome v2.1.1), only *LRP1* variants with high-quality calls (PASS filter value) and with coverage at >10× in >90% of samples were retained for downstream analyses. The rare predicted pathogenic *LRP1* variants were extracted as described above. The total number of alleles evaluated in *LRP1* was taken as the median of the allele numbers recovered for all rare damaging *LRP1* variants as previously described [20,21]. Fisher’s exact test was used to estimate the *p*-value and the odds ratio (OR) with 95% confidence intervals for the Bonferroni-corrected significance threshold [10,22]. Similar burden analyses were conducted for rare synonymous variants in LRP1, which is not expected to be disease-related. This showed a significant increase in burden in cases vs. controls (Appendix A).

## 3. Results 

### 3.1. LRP1 Is Expressed in the Developing Cardiac OFT and AVC

Analyzing published single-cell RNA data in mouse embryos [4] and LRP1 immunostaining (Figure 1A) showed that *Lrp1* is predominantly expressed in OFT and AVC (Figure 1B), consistent with our previous discoveries that *Lrp1* is expressed in the developing AVC, developing ventricle, atria, and OFT in the E10.5 mouse heart [1].

By analyzing two independent published single-cell RNA datasets, we found that *LRP1* is highly expressed in developing human [5] hearts. Spatial transcriptomics of the developing human heart demonstrated *LRP1* is expressed in mesenchymal cells beginning at post-conception week (PCW) 4.5 to 5 and highly expressed in the developing outflow tract (OFT) at PCW 6.5, spanning the time of great arteries development [5] (Figure 2A,B).

### 3.2. LRP1 Is Highly Expressed in the Human Aorta

We used the GTEx database to investigate the transcriptional expression of *LRP1* in a wide range of human adult tissues, including the heart (aorta, ventricle, atria) [23,24]. Examining all the tissues from European-American induvial in the GTEx database demonstrated the highest expression of *LRP1* in the human aorta (Figure 3A). This finding is consistent with the *LRP1* expression in the outflow tract of human [5] and mouse hearts [1,4]. Analysis of scRNA-seq data of human adult aorta [25] demonstrated that LRP1 is highly expressed in fibroblasts, mesothelial, and vascular smooth muscle cells (VSMCs) in the human adult aorta [25] (Figure 3B) consistent with the previous discovery that LRP1 maintains arterial integrity [26].

### 3.3. Nonsynonymous, Rare, and Putative Damaging Variants in LRP1 Are Significantly Associated with CHD

A total of 1922 subjects with CHD (comprised of subjects from PCGC^2^ and CHP) and 2602 control subjects from the Alzheimer’s Disease Sequencing Project (ADSP) (NIAGADS) [3] as a control for background population variation were analyzed. Healthy controls from Alzheimer’s Disease Sequencing Project (ADSP) were treated as the control group in our study. The GnomAD database also employs whole-exome sequencing of healthy samples from ADSP as a GnomAD control subset [3]. In ASDP, cognitively healthy controls were selected with the goal of identifying alleles associated with the increased risk of or protection from late-onset Alzheimer’s disease. All potential controls were at least 60 years old and were either judged to be cognitively normal or did not meet pathological criteria for Alzheimer’s disease following brain autopsy [27]. Further, human exome data from the Exome Aggregation Consortium (ExAC) database of >60,000 individuals [10] with and without CHD showed that a PLI score (indicating the likelihood that a gene is intolerant to a loss of function mutation) and Z score for missense mutation of *LRP1* are 1 and 8.25, respectively. It demonstrates that *LRP1* is highly intolerant to loss-of-function and missense mutations, confirming that this gene is essential for human viability. This shows that rare nonsynonymous variants in *LRP1* affect the cardiac formation and provide important mechanistic insights into gene function and protein domains.

### 3.4. Rare Potentially Pathogenetic Variants (PPV) Are Enriched in CHD

As many missense variants are neutral and would impair the degree of enrichment of pathogenic variants, we filtered in all CHD cohorts (PCGC [2] and CHP) and NIAGADS [3] control cohort by using the following criteria: (1) variant filtering criteria: MAF in genomAD v211 exome <0.0001; (2) for missense variants, it should be deleteriously predicted by at least 4 of 9 prediction algorithms (SIFT, LRT, Polyphen2_HDIV, LRT, MutationTaster, MutationAssessor, FATHMM, PROVEAN, MetaSVM, M_CAP) to be retained for downstream analyses; and (3) loss of function variants were kept for downstream analyses. PPVs in *LRP1* were identified in 58 unrelated individuals with various CHDs. These patients harbor rare PDVs in *LRP1* with a minor allele frequency <0.01% (Figure 4, Appendix A).

Gene burden analysis showed that rare potential damaging missense variants in *LRP1* are the main contributor to CHD rather than loss of function (LoF) (Appendix A). We observed a significant excess of rare damaging *LRP1* variants (missense variants + loss of function) in all CHD (OR = 2.22, *p* = 0.000192), conotruncal defects (CTDs) (OR = 2.37, *p* = 0.00177), left outflow tract obstructions (LVOTO, OR = 1.86, *p* = 0.0307), and AVSD (OR = 3.14, *p* = 0.0194) compared with controls (Figure 5A,B, Appendix A). Similar association results of rare *LRP1* D_Mis variants in CHD were found. We observed that conotruncal defects have the highest odd ratios in relation to rare *LRP1* variants compared with controls. These observations held when we analyzed the rare potentially damaging missense variants in *LRP1* associated with CHD (*p* = 0.000226), CTD (*p* = 0.00309), LVOTO (*p* = 0.0341), and AVSD (*p* = 0.0121) (Appendix A). The results are similar to the rare damaging *LRP1* variants combined with missense variants and loss of function variants (Figure 5, Appendix A). Gene burden analysis also demonstrated a significant excess of rare *LRP1* PDVs. The highest OR (3.24) was observed in Tetralogy of Fallot in line with the observation that *LRP1* is highly expressed in adult aorta across 51 human adult tissue sites from Genotype-Tissue Expression (GTEx) database [23,24] as well as scRNA-seq data [25] (Figure 2 and Figure 3).

### 3.5. Distribution of Rare Damaging Variants in LRP1

Distinct rare putative damaging variants (PDVs) from CHD cohorts mapped along the protein sequence identify putative hotspots of pathogenic damaging mutations. Most PDVs are in the extracellular domain, rarely in the transmembrane or cytoplasmic domains (Figure 4, Appendix A). These damaging variants include a calcium-binding domain (R3014L) and an N-glycosylation site (L4074F). N-glycosylation occurs when sugars are added to the nascent polypeptide chain in the endoplasmic reticulum. After cleavage of glucose and mannose residues, the LRP1 protein is transferred to the Golgi apparatus. Two stop mutations, E2920X and E3802X, are of interest as they remove the C-terminal domain with the NPxY motif essential for clathrin-mediated internalization that would disrupt the endocytic recycling of surface receptors [29]. We note most of the missense mutations in the human genome, including *LRP1*, are heterozygous, indicating they are likely dominant mutations causing gain of function or dominant-negative loss of function. We also identified five patients with compound heterozygous mutations if we used allele frequency <0.01 (Appendix A). In addition, a diverse array of CHD phenotypes is observed with the *LRP1* mutations (Figure 4, Appendix A), likely a reflection of the modifying effects of the genetic background of each patient.

### 3.6. Lrp1 Variants in Patients with AVSD Are Significantly Associated with Their Cases

Previous work has shown that a homozygous missense mutation in *LRP1* in mice results in AVSD and DORV [1]; we sought to determine if a significant relationship exists between *LRP1* variants and AVSD and DORV in the CHP and PCGC population. Of the 1922 patients with CHD, 142 of them have AVSD. The missense variants in these patients are significantly associated with AVSD (*p* = 0.0194). Of the 1922 patients, 111 of them have DORV. There is no significant relationship between variants with an allele frequency under 0.01% and DORV.

## 4. Discussion

Data from projects such as the ExAC provide evidence that rare protein-altering variation is far more common in the general population than we are previously aware of. By using the 2602 healthy controls from the Alzheimer’s Disease Sequencing Project (ADSP) (NIAGADS) as a control for background population variation, we demonstrated a significant association of rare, potentially damaging *LRP1* variants with CHD, especially CTD. The significant enrichment of rare *LRP1* variation in the CHD cohort constitutes evidence of pathogenicity. LRP1 is expressed in the OFT and AVC in developing humans and mice and is highly expressed in the human aorta. The association of rare damaging *LRP1* variants with CHD, especially with conotruncal anomalies, is consistent with the observations of the expression of *LRP1* in the heart at single-cell resolution.

LRP1 is an endocytic trafficking protein. LRP1 is expressed as a 600 kDa precursor cleaved by furin, resulting in a 515 kDa extracellular ligand binding α-chain and a noncovalently bound 85 kDa membrane-bound cytoplasmic β-chain [30]. LRP1 is recognized as a multifunctional receptor that binds to multiple ligands which plays important roles in orchestrating different cellular and molecular aspects: (1) as a scavenger receptor that internalizes multiple extracellular ligands; (2) as a regulatory receptor, it regulates cellular signaling in response to extracellular stimuli; and (3) as a scaffold receptor, LRP1 can partner with and modulate the activity of other membrane proteins such as integrins, bone morphogenetic protein 4 (BMP4), and receptor tyrosine kinases [31]. The importance of *Lrp1* is demonstrated by the early lethality of *Lrp1* gene deletion, as it arrests mouse embryo development at an early stage [32]. The endocytic function and signaling properties confer a major role to LRP1 in the pathophysiology of numerous diseases such as hepatic steatosis, pulmonary hypertension, kidney fibrosis, acute respiratory distress syndrome, Alzheimer’s disease, atherosclerosis, and left ventricular modulation after acute myocardial infarction [31]. We have identified a role of *LRP1* in the pathophysiology of CHD by uncovering a mutant mouse line, 1554 (MGI 96828), that results from a missense (C4232R) mutation in the region encoding the epidermal growth factor (EGF) repeat domain located in the β-chain [1,33]. In this study, we observe an association between rare, potentially damaging *LRP1* variants with human CHDs. These variants are located in different domains/regions of LRP1 protein with potentially damaging interaction with the known pathways associated with CHD pathogenesis, such as BMP4 [34], Notch [35,36,37], and WNT [38,39,40] pathways.

## 5. Conclusions

We reported rare protein-alternating variants in *LRP1* implicated in CHD in a large cohort and identified a significant association with different subtypes of CHD, including CTD, LVOTO, and AVSD. The contribution of *LRP1* rare damaging variants to CHD has the potential diagnostic yield of sequencing that an uncharacterized *LRP1* variant identified in an individual with CHD is pathogenic and informative regarding clinical inheritability of variation in *LRP1*.

## 6. Limitations

The major limitation in this study is that it is only focused on variation in protein-coding or close to protein-coding regions; therefore, we do not fully characterize other variant classes such as non-coding regions, epigenetic, and large structural variants in *LRP1*.

## Figures and Tables

**Figure 1 genes-14-00947-f001:**
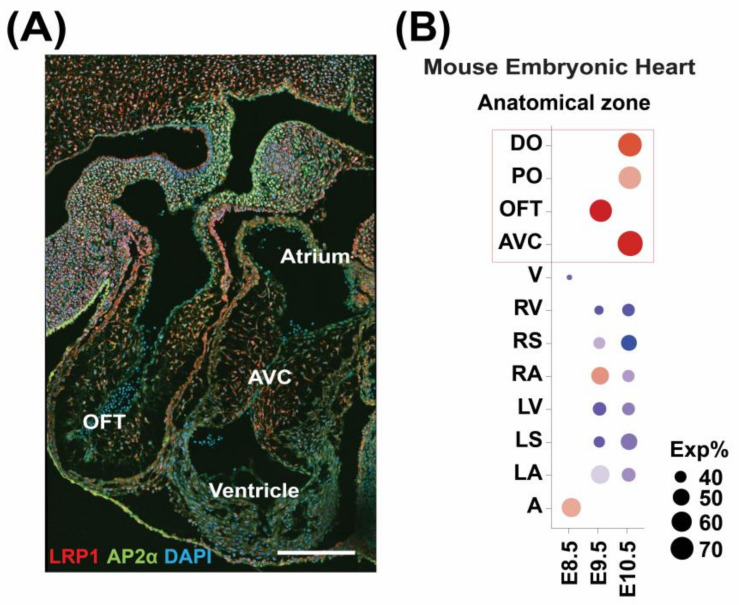
Spatiotemporal expression of *LRP1* in the mouse developing heart. (**A**) Representative LRP1 immunostaining in the E10.5-11.5 mouse heart. AP2α-activating protein transcription factor 2 alpha, a neural crest cell marker. DAPI-4′,6-diamidino-2-phenylindole. Scale bar: 200 microns. (**B**) Expression of *Lrp1* in dissected zones from mouse embryonic heart (from Li et al., 2016, a total of 2407 cells [4]). A-atrium, AVC-atrioventricular cushion, DO-distal outflow tract, LA-left atrium, LS-left ventricular septum, LV-left ventricle, OFT-outflow tract, PO-proximal outflow tract, RA-right atrium, RS-right ventricular septum, RV-right ventricle, V-ventricle.

**Figure 2 genes-14-00947-f002:**
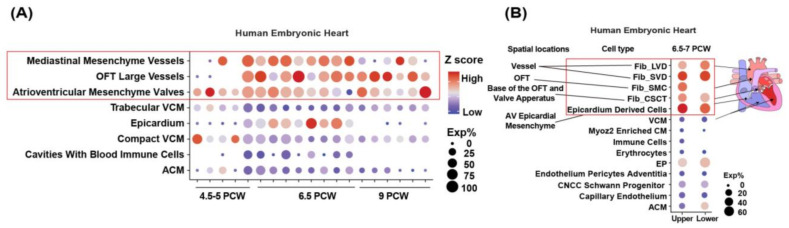
Spatiotemporal expression of LRP1 in the human developing heart. (**A**) Spatiotemporal transcriptional expression of *LRP1* in each cluster in each heart section at each developmental stage (from Asp et al., 2019; a total of 3115 spots [5]). PCW-post-conception weeks. The size of the dot indicates the proportions of cells expressing a certain gene in the respective cell (Exp%), and the color stands for the average expression level of that gene within a cluster in z-scored units at each developmental stage (Z score). (**B**) Cellular expression of *LRP1* in the human embryonic heart at 6.5 PCW (from Asp et al., 2019; a total of 3717 cells [5]). Upper and Lower stands for the upper and lower parts of the heart, respectively. Fib_LVD-Fibroblast-like related to larger vascular development. Fib_SVD-Fibroblast-like related to smaller vascular development. Fib_SMC-Fibroblast-like smooth muscle cell. Fib_CSCT-Fibroblast-like related to cardiac skeleton connective tissue. ACM-atrial cardiomyocyte, AV-atrioventricular, CM-cardiomyocyte, CNCC-cardiac neural crest cell, EP-epicardial cells, OFT-outflow tract, VCM-ventricular cardiomyocyte.

**Figure 3 genes-14-00947-f003:**
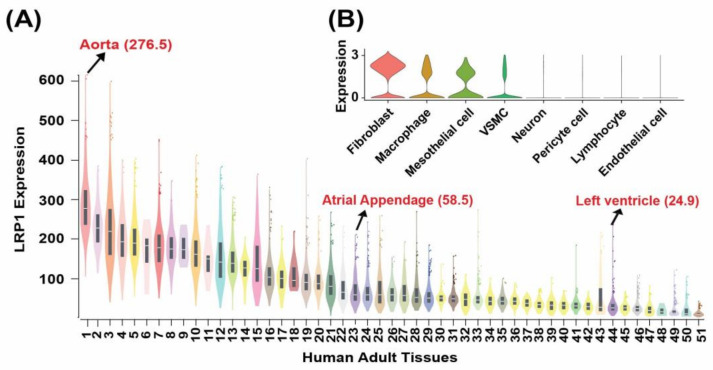
The expression of *LRP1* in adult human tissues. (**A**) Transcriptional expression of *LRP1* in the adult human tissues. The highest transcriptional expression of *LRP1* in the aorta among 54 adult human tissues from the GTEx database. (**B**) Analysis of scRNA-seq data of human adult aorta demonstrated that *LRP1* is highly expressed in fibroblasts, mesothelial, and vascular smooth muscle cells (VSMCs) in the human adult aorta (from Pirruccello et al., 2022; a total of 54,092 cells [25]). (**A**) 1. Aorta, 2. Ovary, 3. Adipose-Subcutaneous, 4. Coronary Artery, 5. Tibial Nerve, 6. Ectocervix, 7. Tibial Artery, 8. Uterus, 9. Endocervix, 10. Omentum (adipose-Visceral), 11. Fallopian Tube, 12. Mammary Tissue, 13. Lung, 14. Cerebellum, 15. Vagina, 16. Esophagus, Gastroesophageal Junction, 17. Cerebellar Hemisphere, 18. Bladder, 19. Esophagus Muscularis, 20. Sigmoid, 21. Thyroid, 22. Prostate, 23. Left Atrial Appendage, 24. Sun-Exposed Skin, 25. Transverse Colon, 26. Minor Salivary Gland, 27. Liver, 28. Terminal Ileum, 29. Not Sun-Exposed Skin, 30. Brain Cortex, 31. Esophagus Mucosa, 32. Nucleus Accumbens, 33. Pituitary, 34. Brain-Caudate, 35. Spleen, 36. Frontal Cortex (BA9), 37. Anterior Cingulate Cortex (BA24), 38. Putamen, 39. Hypothalamus, 40. Amygdala, 41. Adrenal Gland, 42. Substantia Nigra, 43. Stomach, 44. Left Ventricle, 45. Hippocampus, 46. Testis, 47. Spinal Cord (cervical c-1), 48. Kidney Medulla, 49. Skeletal Muscle, 50. Kidney Cortex, 51. Pancreas. GTEx database contains RNA-seq datasets of 54 adult human tissues or organs or cell lines. In Figure 2C, we removed three cell lines and only kept 51 human tissues/organs.

**Figure 4 genes-14-00947-f004:**
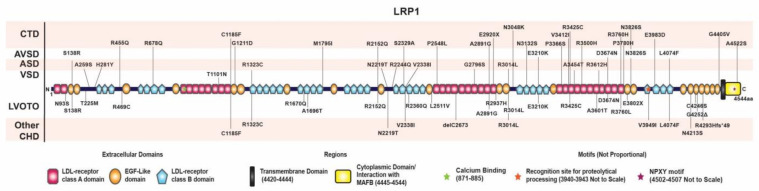
Rare *LRP1* variants are associated with congenital heart defects. Illustration showing the position of rare *LRP1* PDVs found in congenital heart defect (CHD) patients. ASD-atrial septal defect; AVSD-atrioventricular septal defect; CHD-congenital heart defect; CTD-conotruncal defect; LVOTO-left ventricular outflow tract obstruction; VSD-ventricular septal defect.

**Figure 5 genes-14-00947-f005:**
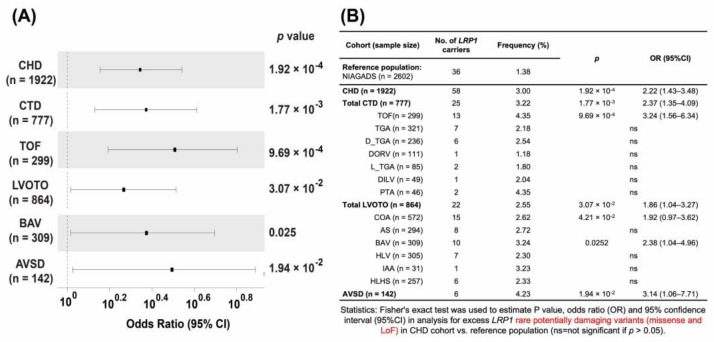
Rare *LRP1* damaging variants are associated with different congenital heart defects. (**A**) Compared with normal controls, *LRP1* damaging variants are associated with different CHDs, including conotruncal defect (CTD), Tetralogy of Fallot (TOF), left outflow tract obstruction (LVOTO), bicuspid aortic valve (BAV), and atrioventricular septal defect (AVSD). The odd ratio (OR) and 95% confidence interval (CI) are shown. The total number of affected individuals analyzed is indicated. (**B**) The statistical analysis compared with LRP1 rare potential damaging variants with different congenital heart defects using the normal control from ADSP/NIAGADS [28] as a control reference.

## Data Availability

The LRP1 variants identified in Pittsburgh cohort are available under SRA numbers BioProject accession number PRJNA632119. Publicly available single-cell RNA (scRNA-seq) sequencing dataset of fetal mouse heart at E8.5, E9.5, and E10.5 were downloaded from the NCBI/GEO database (accession no GSE76118). Single nucleus RNA sequencing data of adult human aorta are publicly available at the Broad Institute’s Single Cell Portal (accession no. SCP1265). scRNA-seq dataset and Spatial transcriptomics of fetal human heart at 4.5–5, 6.5, and 9 post-conception weeks were downloaded from https://www.spatialresearch.org/. Whole exome sequencing data of samples from PCGC and ADSP were available from NCBI dbGaP database (accession no phs001194.v2.p2) and through NIAGADS DSS (NG00067.v2) with permission, respectively.

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
