# Peer review of "Contribution of LRP1 in Human Congenital Heart Disease Correlates with Its Roles in the Outflow Tract and Atrioventricular Cushion Development"

_genes, 2023, doi:10.3390/genes14040947_

Round 1

Reviewer 1 Report

The manuscript should be published as submitted to the editor. Should the "material and methods" chapter be after the "discussion" chapter?

Author Response

 Reviewer #1 

The manuscript should be published as submitted to the editor. Should the "material and methods" chapter be after the "discussion" chapter? 

Answer: Thank you very much for your suggestion, we made the changes according to your request. 

Reviewer 2 Report

Excellent Abstract and introduction to set the flow. 

Please consider including demographics data for the CHD and Control group including the age group and compare it within the discussion section.

Overall excellent approach and data presentation. 

No further major comments.

Author Response

 Reviewer #2 

Please consider including demographics data for the CHD and Control group including the age group and compare it within the discussion section. 

A: We were unable to the information regarding the exact dates and ages of blood sampling for the whole exome sequence. However, we do provide the demographic information of patients’ sexes, races, major congenital heart defects, and extracardiac lesions. We use healthy controls without cognitive anomalies from Alzheimer's Disease Sequencing Project (ADSP) as the control group in our study. These controls are all adults, they are much older than our pediatric congenital heart disease cohort. 

Reviewer 3 Report

Among birth defects, congenital heart disease is the leading cause of infant mortality, affecting approximately 7-10 out of 1000 live births. CHD encompasses a broad spectrum of cardiac malformations, ranging from a single anomaly to complex lesions composed of multiple defects. Genetic etiologies contribute to an estimated 90% of CHD cases, so far, however, a molecular diagnosis remains unresolved in the majority of the patients.

LRP1 is a member of the LDL receptor gene family. To date, LRP1 is known to bind to more than 40 distinct ligands that are structurally unrelated. LRP1 is expressed in multiple cell types.  The large number and functionally diverse ligands for LRP1 and the embryonic lethality in global LRP1-/- mice suggest that LRP1 is essential in various physiological and pathological processes.

 The submitted manuscript of A. Arrigo et al. builds on an earlier work from this research group, in which they were able to show for the first time an influence of LRP1 on the development of the heart in a mouse model. Using RNA databases, they have now attempted to refine the local expression of LRP1 in the developing heart in both mice and humans. In addition, they used the GTEx database to look for the expression profile in different human adult tissues. Finally, the authors examined the frequency of LRP1 mutations in two CHD cohorts (n= 1922) and in a control group (n= 2602) and their influence on the different manifestations of CHD. They identified potentially pathogenetic variants in 58 unrelated individuals in the CHD cohort.

The reviewer would like to make the following comments:

Figure 1 is overloaded and even if I understand the intention of the authors to want to show mouse and human opposite, I would prefer a separation.

I don't understand why figure 2 combines LRP1 expression data of human tissues and data of LRP1 genetic variants and the association of these variants with different forms of CHD.

The number of the potentially pathogenetic variants which is the basis for the statistical analysis of Figure 2 should not be hidden in the supplement part.

I'm not sure if the decision to use an Alzheimer's collective as a control group is correct. This group is older and is likely to contain fewer people with heart damage per se than another control group, since they may have died by now.

CHD is a heterogeneous disease that can be caused by different genes. In the 58 described carriers of LRP1 mutations, has it been ruled out beforehand that there can be no other genetic cause?

LRP1 is a very large protein and it is therefore difficult to prove that a mutation is pathogenic. I acknowledge the authors' attempt to eliminate this weak point by using various prediction programs, but I would like a more critical appraisal in the discussion. In this context, the question arises whether the CHD phenotype of the person in the CDH cohort also occurs in the allele-carrying parent or sibling.

Author Response

 Reviewer #3 

-Figure 1 is overloaded and even if I understand the intention of the authors to want to show the mouse and human opposite, I will prefer a separation. 

Answer: We made changes according to your comments. We separated the mouse and human data. 

-I don't understand why figure 2 combines LRP1 expression data of human tissues and data of LRP1 genetic variants and the association of these variants with different forms of CHD. 

Answer: We separated the LRP1 expression data of human tissues and data of LRP1 genetic variants. 

-The number of the potential pathogenetic variants which is the basis for the statistical analysis of Figure 2 should not be hidden in the supplement part. 

Answer: Thank you very much for your suggestion, we made the changes according to your request. 

-I'm not sure if the decision to use an Alzheimer's collective as a control group is correct. This group is older and is likely to contain fewer people with heart damage per se than another control group, since they may have died by now. 

Answer: Healthy controls from Alzheimer's Disease Sequencing Project (ADSP) were treated as the control group in our study. The GnomAD database also employs whole exome sequencing of healthy samples from ADSP as a GnomAD control subset. And, In ASDP, cognitively healthy controls were selected with the goal of identifying alleles associated with the increased risk of or protection from late-onset AD. At the time of the last exam, all potential controls were at least 60 years old and were either judged to be cognitively normal or did not meet pathological criteria for Alzheimer’s disease following brain autopsy1. It is highly likely that congenital heart diseases in these elder people were detected and recorded by the time of the exam. 

-CHD is a heterogeneous disease that can be caused by different genes. In the 58 described carriers of LRP1 mutations, has it been ruled out beforehand that there can be no other genetic cause? 

LRP1 is a very large protein, and it is therefore difficult to prove that a mutation is pathogenic. I acknowledge the authors' attempt to eliminate this weak point by using various prediction programs, but I would like a more critical appraisal of the discussion. In this context, the question arises whether the CHD phenotype of the person in the CDH cohort also occurs in the allele-carrying parent or sibling. 

Answer: They are tough questions. More than 400 susceptibility genes had been reported to contrite to CHD. Potentially damaging mutations in some genes among these 400 genes also could be found in these 58 described carriers of LRP1 mutations. However, in our study, we conducted a gene-burden association analysis to investigate whether potentially damaging LRP1 mutations are enriched in CHD types, indicating LRP1 mutations have an effect on CHD. 

Reference 

1. Bis, J.C., et al. Whole exome sequencing study identifies novel rare and common Alzheimer’s-Associated variants involved in immune response and transcriptional regulation. Molecular Psychiatry 25, 1859-1875 (2020). 
